# MULTITASK PROMPT TUNING ENABLES PARAMETER-EFFICIENT TRANSFER LEARNING

**Zhen Wang**[1]* **Rameswar Panda**[2] **Leonid Karlinsky**[2] **Rogerio Feris**[2] **Huan Sun**[1] **Yoon Kim**[3]

[1]The Ohio State University, [2]MIT-IBM Watson AI Lab, [3]Massachusetts Institute of Technology
{wang.9215,sun.397}@osu.edu, {rpanda, leonidka, rsferis}@ibm.com, yoonkim@mit.edu

## ABSTRACT

Prompt tuning, in which a base pretrained model is adapted to each task via conditioning on learned prompt vectors, has emerged as a promising approach for efficiently adapting large language models to multiple downstream tasks. However, existing methods typically learn soft prompt vectors from scratch, and it has not been clear how to exploit the rich cross-task knowledge with prompt vectors in a multitask learning setting. We propose multitask prompt tuning (MPT), which first learns a single transferable prompt by distilling knowledge from multiple task-specific source prompts. We then learn multiplicative low rank updates to this shared prompt to efficiently adapt it to each downstream target task. Extensive experiments on 23 NLP datasets demonstrate that our proposed approach outperforms the state-of-the-art methods, including the full finetuning baseline in some cases, despite only tuning $0.035\%$ as many task-specific parameters.[1]

## 1 INTRODUCTION

Finetuning pretrained language models (PLMs) has led to significant improvements across various downstream NLP tasks (Devlin et al., 2019; Howard & Ruder, 2018; Raffel et al., 2020). However, the conventional paradigm of full task-specific finetuning (FT) is difficult to scale to multiple tasks, given that modern PLMs can have hundreds of millions (or even billions) of parameters. There thus has been a growing interest in developing *parameter-efficient* methods for model tuning (Houlsby et al., 2019; Lester et al., 2021; Ding et al., 2022), where the goal is to learn only a small number of additional parameters per task while achieving performance comparable to full finetuning.

Prompt tuning (PT), which prepends tunable continuous prompt vectors to the input, has emerged as a promising approach for parameter-efficient transfer learning with PLMs (Liu et al., 2021a; Li & Liang, 2021; Lester et al., 2021; Liu et al., 2022b; 2021b). PT freezes the PLM parameters and only learns a small set of task-specific prompt vectors. However, despite their impressive performance, there is still a large gap between prompt tuning and full finetuning (Lester et al., 2021). Additionally, this approach is sensitive to initialization and often requires more training time than finetuning (Su et al., 2022; Zhong et al., 2022).

Recent work has proposed to address these issues by *transferring* prompt vectors from various tasks (Su et al., 2022; Zhong et al., 2022). These methods first train soft prompts on multiple source tasks and then use these pretrained prompts to initialize the prompt for further finetuning on a target task based on a (potentially learned) similarity measure (Vu et al., 2022; Asai et al.,

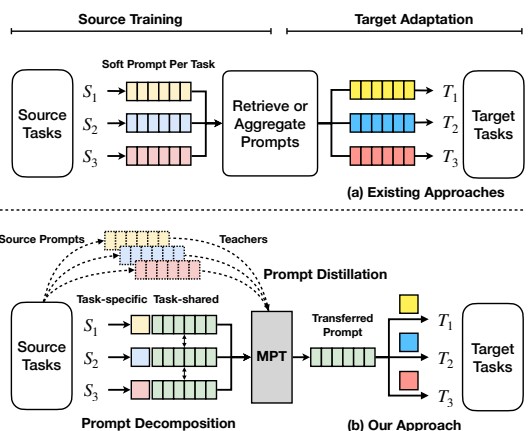

**Figure 1:** A conceptual overview of our approach. Instead of retrieving or aggregating source prompts (top), multitask prompt tuning (MPT, bottom) learns a single transferable prompt. The transferable prompt is learned via prompt decomposition and distillation.

---

*Work done during an internship at MIT-IBM Watson AI Lab.
[1]Project page: https://zhenwang9102.github.io/mpt.html

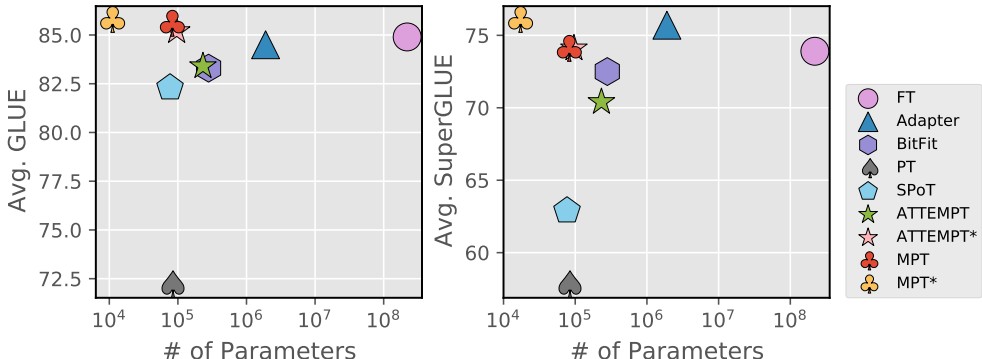

**Figure 2:** Parameter efficiency on GLUE (left) and SuperGLUE (right). Our multitask prompt tuning (MPT) approach, which transfers a single shared prompt learned from multiple source tasks using prompt decomposition and distillation, maintains high accuracy ($y$-axis) while finetuning only a small number of parameters per task ($x$-axis). All results are based on T5-Base (Raffel et al., 2020). Baselines include: Adapters (Houlsby et al., 2019), BitFit (Zaken et al., 2022), PT (Lester et al., 2021), SPoT (Vu et al., 2022), and ATTEMPT (Asai et al., 2022). *Indicates multitask training on target tasks. Best viewed in color.

2022) (see Figure 1, top). In this paper, we extend this line of work and introduce *multitask prompt tuning* (MPT), which uses multitask data to learn a *single* prompt that can be efficiently transferred to target tasks. While conceptually simple, learning a shared prompt space can be practically challenging as it requires learning commonalities across different source tasks while minimizing interference. Therefore, we decompose the soft prompt of each source task (which can be represented as a prompt matrix) into a multiplication of a shared matrix and a low-rank task-specific matrix, and find that this decomposition is more effective than simply sharing the prompt matrix across all tasks. This decomposition is learned through knowledge distillation from soft prompts obtained from regular prompt tuning. To transfer to new tasks, we perform low-rank multiplicative updates to the shared prompt matrix. Figure 1 (bottom) illustrates our approach.

Extensive experiments on 23 NLP datasets across diverse tasks demonstrate the effectiveness of our proposed approach over state-of-the-art prompt transfer methods. On the SuperGLUE benchmark (Wang et al., 2019), MPT with T5-Base (Raffel et al., 2020) yields a $16.3\%$ improvement over the vanilla prompt tuning baseline (PT, Lester et al., 2021), and also outperforms the most competitive multitask prompt transfer baseline (ATTEMPT, Asai et al., 2022) despite tuning much fewer task-specific prompt parameters (77.6K vs 232K). On some benchmarks, MPT exceeds the performance of full finetuning while only requiring $0.035\%$ tunable parameters per task (see Figure 2). We also find that MPT is very effective for few-shot learning with 4-32 labels for each target task.

## 2 RELATED WORK

**Parameter-efficient transfer learning.** Parameter-efficient transfer learning for pretrained language models is an active research area (Ding et al., 2022). Adapters (Houlsby et al., 2019; Mahabadi et al., 2021) and its variants (Hu et al., 2021; Karimi Mahabadi et al., 2021) insert trainable layers, while BitFit (Zaken et al., 2022) only updates the bias parameters without changing any other model parameters. Diff pruning (Guo et al., 2021) and FISH (Sung et al., 2021) learn sparse updates to the original PLM. Another popular choice is prompt tuning (Lester et al., 2021) which only updates soft prompt vectors prepended to the input. Prefix-tuning of optimizing continuous prompts for natural language generation tasks is presented in Li & Liang (2021). UNIPELT learns to combine different tuning methods via gating mechanism (Mao et al., 2022). HyperPrompt (He et al., 2022) introduces task-conditioned hyperprompts that condition the model on task-specific information for constructing prompts. LST (Sung et al.) aims to reduce the training memory of parameter-efficient tuning by a ladder side network. Discrete (i.e., hard) prompts have also been shown to be effective in many cases (Schick & Schütze, 2021a;b; Gao et al., 2021; Malkin et al., 2022). However, our approach is most related to the transferability of prompts (Wang et al., 2021; Vu et al., 2022; Su et al., 2022), which focuses on boosting the performance of prompt tuning across many tasks. SPoT (Vu et al., 2022) selects one prompt using a similarity measure, and ATTEMPT (Asai et al., 2022) adopts an attention mechanism over the source prompts to initialize the prompt for a target task. Unlike existing works, our approach learns a single shared prompt by decomposing and distilling knowledge from source prompts for efficient adaptation to a diverse set of target tasks.

**Multitask learning.** Multitask learning, which focuses on simultaneously solving multiple related tasks with a single model, has been studied from multiple perspectives (Zhang & Yang, 2021; Ruder, 2017). A common approach is to transfer a model that has been fine-tuned on multiple source tasks to another target task (Vu et al., 2020; Raffel et al., 2020; Aghajanyan et al., 2021a; Zhong et al., 2021; Clark et al., 2019b; Singh et al., 2022). A few recent works show zero-shot and few-shot transfer capabilities of language models through massive multitask learning over a large number of tasks (Sanh et al., 2022; Wang et al., 2022; Liu et al., 2022a; Wei et al., 2021). Designing specific parameter-sharing strategies is also another recent trend in multitask learning (Ruder et al., 2019; Sun et al., 2020; Misra et al., 2016). While our proposed approach is inspired by these methods, this paper focuses on multitask prompt transfer for parameter-efficient adaptation of language models, which still remains a challenging and largely understudied problem.

**Knowledge distillation.** Knowledge distillation has been used to improve performance and efficiency across many tasks (Gou et al., 2021), including model compression (Hinton et al., 2015; Jiao et al., 2020; Sanh et al., 2019), transfer learning (Furlanello et al., 2018; Xu et al., 2020), machine translation (Zhou et al., 2019), question answering (Hu et al., 2018), and document retrieval (Shakeri et al., 2019). Concurrently with our work, PANDA (Zhong et al., 2022) uses knowledge distillation with a new metric to better predict prompt transferability across different combinations of source-target tasks. PANDA focuses on transferring from one source task to another target task using a similarity measure (similar to SPoT (Vu et al., 2022)), while our MPT approach leverages multitask learning to better exploit cross-task knowledge for prompt transfer.

## 3 APPROACH

Given a set of source tasks $\boldsymbol{\mathcal{S}} = \{\mathcal{S}_1, \mathcal{S}_2, ..., \mathcal{S}_\kappa\}$ and target tasks $\boldsymbol{\mathcal{T}} = \{\mathcal{T}_1, \mathcal{T}_2, ..., \mathcal{T}_\tau\}$, our goal is to learn a single soft prompt over $\boldsymbol{\mathcal{S}}$ that can be adapted to each task $\mathcal{T}_i$ in a parameter-efficient way. Simply training a single soft prompt on $\boldsymbol{\mathcal{S}}$ and then finetuning on each $\mathcal{T}_i$ is sub-optimal as it can fail to leverage commonalities across source tasks while minimizing interference at the same time. To this end, multitask prompt tuning (MPT) aims to compress task-shared knowledge in $\boldsymbol{\mathcal{S}}$ into a single prompt matrix $\phi_{\boldsymbol{\mathcal{S}}}$ via knowledge distillation to improve performance on $\boldsymbol{\mathcal{T}}$ while filtering out task-specific information that is less useful for transfer learning.

**Prompt tuning.** Given a pre-trained language model with parameters $\Theta$ and one target task $\mathcal{T}$ with training data $(\boldsymbol{X}, \boldsymbol{Y}) = \{\boldsymbol{x}_i, \boldsymbol{y}_i\}_{i=1}^N$, the standard approach is to directly finetune all the parameters by maximizing the conditional probability $P(\boldsymbol{Y}|\boldsymbol{X}; \Theta)$, which can be parameter-inefficient when considering a group of target tasks $\boldsymbol{\mathcal{T}}$. An alternative that is more parameter-efficient is prompt tuning (PT), which randomly initializes a small number of learnable prompt vectors (i.e., soft prompts) to be prepended to the input embeddings of the PLM while freezing model parameters $\Theta$ (Lester et al., 2021; Liu et al., 2022b). Formally, for a sequence of input tokens with token embeddings as $\boldsymbol{T} = [\boldsymbol{t}_1, \boldsymbol{t}_2, ..., \boldsymbol{t}_n] \in \mathbb{R}^{n \times d}$, PT prepends a learnable prompt matrix $\boldsymbol{P} \in \mathbb{R}^{l \times d}$ with the same dimension as the token embedding $d$, where $l$ is a hyperparameter. PT then optimizes the following loss function with respect to $\boldsymbol{P}$,

$$\mathcal{L}_{\text{PLM}} = -\sum_i \log P(\boldsymbol{y}_i \,|\, \boldsymbol{x}_i \,; \Theta, \boldsymbol{P}), \tag{1}$$

where the input to the language model is given by the concatenated matrix $[\boldsymbol{P}; \boldsymbol{T}] \in \mathbb{R}^{(l+n) \times d}$. While this approach has been successful for some tasks and models, researchers have observed that vanilla PT can sometimes lead to lower performance (especially on smaller PLMs), slow convergence, and high sensitivity to parameter initialization (Lester et al., 2021; Su et al., 2022; Zhong et al., 2022). Recent works address these issues by first training prompts on multiple source tasks and then using these prompts to initialize the prompts for a target task via some similarity measure (Asai et al., 2022; Vu et al., 2022). We extend this line of work and propose a framework for transferring multitask knowledge into a single soft prompt to enable more performant and parameter-efficient transfer learning to downstream target tasks $\boldsymbol{\mathcal{T}}$.

### 3.1 MULTITASK PROMPT TUNING

Our proposed framework, dubbed MPT, consists of two stages: *source training* and *target adaptation*. MPT first focuses on source training to generate a *single* soft prompt matrix to be reused in the

second stage for target task adaptation. Specifically, prompt matrices for the source tasks are decomposed into a task-shared matrix and a low-rank task-specific matrix (*prompt decomposition*), where the former is shared across all tasks. This decomposition into shared and task-specific components is learned through knowledge distillation. Once learned, the shared prompt matrix is adapted to a downstream target task via low-rank multiplicative updates.

**Prompt decomposition.** The goal of prompt decomposition is to enable efficient knowledge sharing across source tasks $\boldsymbol{\mathcal{S}}$, while still allowing each task to maintain its own parameters to encode task-specific knowledge. We decompose the soft prompt $\boldsymbol{P}_k$ for the $k$-th task into two parts, as shown in Figure 3. Let $\boldsymbol{P}^* \in \mathbb{R}^{l \times d}$ denote the shared prompt across all tasks, and further let $\boldsymbol{u}_k \in \mathbb{R}^l, \boldsymbol{v}_k \in \mathbb{R}^d$ be the task-specific vectors for each task $k$. The task-specific vectors form a rank-one matrix $\boldsymbol{W}_k = \boldsymbol{u}_k \otimes \boldsymbol{v}_k^T$, which has the same dimensions as the shared prompt $\boldsymbol{P}^*$. The task prompt $\widehat{\boldsymbol{P}}$ for $k$-th source task is then parameterized as:

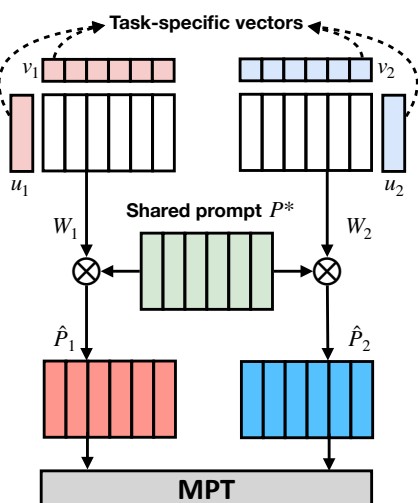

$$\widehat{\boldsymbol{P}}_k = \boldsymbol{P}^* \circ \boldsymbol{W}_k = \boldsymbol{P}^* \circ (\boldsymbol{u}_k \otimes \boldsymbol{v}_k^T), \quad (2)$$

where $\circ$ denotes the Hadamard product between two matrices. Our parameterization of prompt decomposition is inspired by prior low-rank methods (Li et al., 2018; Aghajanyan et al., 2021b; Wen et al., 2020), such that general information across the set of source tasks $\boldsymbol{\mathcal{S}}$ can be captured by "slow" weights $\boldsymbol{P}^*$ shared across tasks, while the "fast" weights $\boldsymbol{W}_k$ could then encode task-specific knowledge for $\mathcal{S}_k$ in a low-rank subspace.

**Figure 3:** An illustration on prompt decomposition for two tasks. The shared matrix $\boldsymbol{P}^\star$ is combined with task-specific vectors $\boldsymbol{u}_k, \boldsymbol{v}_k$ to obtain the task-specific prompt matrices $\widehat{\boldsymbol{P}}_k$ for $k \in \{1, 2\}$.

**Prompt distillation.** Learning the prompt decomposition directly from the multitask datasets $\boldsymbol{\mathcal{S}}$ tended to make the shared component $\boldsymbol{P}^*$ overfit to larger tasks. We found knowledge distillation from separately-trained source prompts to be an effective strategy for learning good decomposable prompts. Specifically, we first obtain a teacher prompt $\boldsymbol{P}_k^{(teacher)}$ for the $k$-th source task by conventional prompt tuning. We then randomly initialize a corresponding student prompt as $\widehat{\boldsymbol{P}}_k = \boldsymbol{P}^* \circ (\boldsymbol{u}_k \otimes \boldsymbol{v}_k^T)$, where all student prompts share $\boldsymbol{P}^*$ and have their own task-specific vectors as described above. We then use distillation to transfer cross-task knowledge into the shared prompt matrix (Sanh et al., 2019). The first loss is to match the output probability distributions of students and teachers by minimizing their KL-Divergence with respect to the shared prompt matrix $\boldsymbol{P}^*$ and the task-specific parameters $\boldsymbol{u}_k$ and $\boldsymbol{v}_k$,

$$\mathcal{L}_{\text{Logits}} = \sum_{k \in |\boldsymbol{\mathcal{S}}|} \sum_{(\boldsymbol{x}_i, \boldsymbol{y}_i) \in \mathcal{S}_k} \text{KL} \left[ P\left( \boldsymbol{y}_i \,|\, \boldsymbol{x}_i \,;\, \Theta, \boldsymbol{P}_k^{(teacher)} \right) \,\|\, P\left( \boldsymbol{y}_i \,|\, \boldsymbol{x}_i \,;\, \Theta, \widehat{\boldsymbol{P}}_k \right) \right]. \quad (3)$$

We use a temperature $T$ to control the smoothness of the output distribution for both teacher and student models as $p_j = \frac{1}{Z} \exp(z_j / T)$, where $z_i$ is the logit score for class $j$ and $Z$ is the normalization factor. We also have an additional mean squared loss on teacher model hidden states,

$$\mathcal{L}_{\text{Hidden}} = \sum_{k \in |\boldsymbol{\mathcal{S}}|} \sum_{(\boldsymbol{x}_i, \boldsymbol{y}_i) \in \mathcal{S}_k} (\boldsymbol{H}_{k,i} - \boldsymbol{H}_{k,i}^{(teacher)})^2, \quad (4)$$

where $\boldsymbol{H}_{k,i}^{(teacher)}$ and $\boldsymbol{H}_{k,i}$ denote the hidden states of teacher and student networks respectively, which consist of a sequence of hidden vectors for $i$-th input. Such additional distillation loss from intermediate states has been shown to improve results in distilling PLMs (Jiao et al., 2020; Shleifer & Rush, 2020). The total loss function for training student source prompts for obtaining a single shared prompt to be transferred to the target side is then,

$$\mathcal{L}_{\text{Total}} = \mathcal{L}_{\text{PLM}} + \lambda(\mathcal{L}_{\text{Logits}} + \mathcal{L}_{\text{Hidden}}), \quad (5)$$

where $\mathcal{L}_{\text{PLM}} = \sum_{k \in |\boldsymbol{\mathcal{S}}|} \mathcal{L}_{\text{PLM}}^k$ represents the aggregated task losses for all source tasks, and $\lambda$ is a weight to balance the impact of distillation loss terms.

## 3.2 SOURCE TRAINING AND TARGET ADAPTATION

Training the single source prompt to be transferred to target tasks requires two steps. First, the teacher prompts for all source tasks are pretrained individually through vanilla prompt tuning. Then, we perform multitask training on $\boldsymbol{\mathcal{S}} = \{\mathcal{S}_1, \ldots, \mathcal{S}_\kappa\}$ to jointly learn the single shared prompt via the knowledge distillation loss function in Equation 5. We also adopt a simple stochastic task sampling strategy, which dynamically changes the number of tasks per batch. For each batch of multitask samples, we randomly select a number $K$ from $[2, \kappa]$ first, then randomly choose $K$ tasks from $\boldsymbol{\mathcal{S}}$ and their corresponding samples to constitute mini-batches. Such dynamic task sampling strategies are common in the PLM multitask learning literature (Raffel et al., 2020).

For target adaptation, we initialize the target prompt for target task $\mathcal{T}_t$ to be the Hadamard product of the shared prompt matrix and the task-specific low-rank prompt matrix, i.e., $\widehat{\boldsymbol{P}}_t = \boldsymbol{P}^* \circ (\boldsymbol{u}_t \otimes \boldsymbol{v}_t^\top)$ and optimize with the regular task loss in Equation 1 with respect to $\boldsymbol{P}^*, \boldsymbol{u}_t, \boldsymbol{v}_t$, where we use separate learning rates for $\boldsymbol{P}^*$ vs. $\boldsymbol{u}_t, \boldsymbol{v}_t$ (see Appedix A). We remark that MPT can also be used for multitask learning on a *group* of target tasks $\boldsymbol{\mathcal{T}} = \{\mathcal{T}_1, \mathcal{T}_2, ..., \mathcal{T}_\tau\}$, where $\boldsymbol{P}^*$ is shared across $\boldsymbol{\mathcal{T}}$.

**Parameter-efficiency.** Each task contains the shared prompt $l \times d$ that has the same dimensions as a vanilla soft prompt and a smaller number of task-specific vectors $(l + d)$. Thus, the total number of tunable parameters for a single target task is $(l \times d) + (l + d)$. After training, this can further be compressed into a single matrix of size $l \times d$.[2] For a *group* of target tasks, the total number of tunable parameters is $(l \times d) + (l + d)\tau$, where $\tau$ is the number of target tasks. We list and compare different methods in terms of the number of trainable parameters in Table 1.

## 4 EXPERIMENTS

We conduct experiments across a comprehensive range of NLP datasets to show that MPT outperforms strong baselines in both full-dataset (Tables 1, 2) and few-shot (Tables 3, 4) adaptations, while being more parameter-efficient compared to existing methods (Figure 2).

## 4.1 EXPERIMENTAL SETUP

**Datasets and tasks.** As in Asai et al. (2022) we evaluate MPT using 6 datasets with more than 100k annotations as *source* tasks: MNLI (Williams et al., 2017), QNLI (Demszky et al., 2018), QQP (Wang et al., 2018), SST-2 (Socher et al., 2013), SQuAD (Rajpurkar et al., 2016), and ReCoRD (Zhang et al., 2018). We use 23 datasets from four benchmarks as *target* tasks: MultiRC (Khashabi et al., 2018), BoolQ (Clark et al., 2019a), WiC (Pilehvar & Camacho-Collados, 2018), WSC (Levesque et al., 2012), and CB (De Marneffe et al., 2019) from SuperGLUE (Wang et al., 2019); RTE (Giampiccolo et al., 2007), CoLA (Warstadt et al., 2019), STS-B (Cer et al., 2017), MRPC (Dolan & Brockett, 2005), MNLI, QQP, QNLI and SST-2 from GLUE (Wang et al., 2018); Natural Questions (Kwiatkowski et al., 2019), HotpotQA (Yang et al., 2018), NewsQA (Trischler et al., 2017) and SearchQA (Dunn et al., 2017) from MRQA (Fisch et al., 2019); WinoGrande (Sakaguchi et al., 2021), Yelp-2 (Zhang et al., 2015), SciTail (Khot et al., 2018) and PAWS-Wiki (Zhang et al., 2019) from the "Others" benchmark in (Asai et al., 2022); and E2E (Novikova et al., 2017) and WebNLG (Gardent et al., 2017) for experiments on adapting to natural language generation tasks.

**Models.** Following the standard approach in prompt tuning (Lester et al., 2021; Asai et al., 2022), we mainly experiment using the publicly available pretrained T5-Base model with 220M parameters (Raffel et al., 2020). We use 100 prompt vectors for all benchmarks (hence $\widehat{\boldsymbol{P}}_k \in \mathbb{R}^{100 \times d}$). In our ablation study, we also consider T5-Small (60M) and T5-Large (770M) models.

**Baselines.** We compare MPT with the following baselines: (1) Full finetuning (FT), where all the model parameters are tuned during adaptation on each downstream task. (2) Vanilla prompt tuning (PT) (Lester et al., 2021), where target prompt vectors are initialized by randomly sampled top vocabularies. (3) Existing prompt transfer methods, including SPoT (Vu et al., 2022) and ATTEMPT (Asai et al., 2022), which initialize target prompts by retrieving or aggregating source prompts. (4) Popular parameter-efficient methods including Adapters (Houlsby et al., 2019) and BitFit (Zaken et al., 2022). On GLUE, we also compare with several state-of-the-art methods that

---

[2]However for comparison against prior work we show the number of tunable parameters, i.e., $(l \times d) + (l + d)$.

**Table 1:** Results on GLUE and SuperGLUE. The metrics are Pearson correlation for STS-B, F1 for MultiRC (Multi), and accuracy for other tasks as evaluation metrics. MPT results are averaged over three runs, and subscripts denote standard deviation. The column "param/task" represents the number of trainable parameters for each task in GLUE. (Top) Model adaptation to each target task with no parameter sharing on the target side (so params/task for MPT is just $(l \times d) + (l + d)$). (Bottom) Model adaptation to a *group* of tasks (marked by *), where param/task for MPT * is $(l \times d)/\tau + (l + d)$. See Section 3.2 for more details.

| Method | param/task | GLUE | | | | | | | | | SuperGLUE | | | | | |
| --- | --- | --- | --- | --- | --- | --- | --- | --- | --- | --- | --- | --- | --- | --- | --- | --- |
| | | MNLI | QQP | QNLI | SST-2 | STS-B | MRPC | RTE | CoLA | Avg. | Multi | BoolQ | WiC | WSC | CB | Avg. |
| Finetuning | 220M | 86.8 | 91.6 | 93.0 | 94.6 | 89.7 | 90.2 | 71.9 | 61.8 | 84.9 | 72.8 | 81.1 | 70.2 | 59.6 | 85.7 | 73.9 |
| Adapters | 1.9M | 86.5 | 90.2 | 93.2 | 93.8 | 90.7 | 85.3 | 71.9 | 64.0 | 84.5 | 75.9 | 82.5 | 67.1 | 67.3 | 85.7 | 75.7 |
| BitFit | 280K | 85.3 | 90.1 | 93.0 | 94.2 | 90.9 | 86.8 | 67.6 | 58.2 | 83.3 | 74.5 | 79.6 | 70.0 | 59.6 | 78.6 | 72.5 |
| PT | 76.8K | 81.3 | 89.7 | 92.8 | 90.9 | 89.5 | 68.1 | 54.7 | 10.6 | 72.2 | 58.7 | 61.7 | 48.9 | 51.9 | 67.9 | 57.8 |
| SPoT | 76.8K | 85.4 | 90.1 | 93.0 | 93.4 | 90.0 | 79.7 | 69.8 | 57.1 | 82.3 | 74.0 | 77.2 | 67.0 | 50.0 | 46.4 | 62.9 |
| ATTEMPT | 232K | 84.3 | 90.3 | 93.0 | 93.2 | 89.7 | 85.7 | 73.4 | 57.4 | 83.4 | 74.4 | 78.8 | 66.8 | 53.8 | 78.6 | 70.5 |
| MPT | 77.6K | $85.9_{0.07}$ | $90.3_{0.00}$ | $93.1_{0.07}$ | $93.8_{0.09}$ | $90.4_{0.05}$ | $89.1_{0.23}$ | $79.4_{1.22}$ | $62.4_{0.94}$ | $\mathbf{85.6}_{0.33}$ | $74.8_{0.07}$ | $79.6_{0.43}$ | $69.0_{0.25}$ | $67.3_{0.00}$ | $79.8_{2.91}$ | $\mathbf{74.1}_{0.73}$ |
| Finetuning* | 28M | 85.7 | 91.1 | 92.0 | 92.5 | 88.8 | 90.2 | 75.4 | 54.9 | 83.8 | - | - | - | - | - | - |
| Adapters* | 1.8M | 86.3 | 90.5 | 93.2 | 93.0 | 89.9 | 90.2 | 70.3 | 61.5 | 84.4 | - | - | - | - | - | - |
| HyperFomer* | 638K | 85.7 | 90.0 | 93.0 | 94.0 | 89.7 | 87.2 | 75.4 | 63.7 | 84.8 | - | - | - | - | - | - |
| HyperDecoder* | 1.8M | 86.0 | 90.5 | 93.4 | 94.0 | 90.5 | 87.7 | 71.7 | 55.9 | 83.7 | - | - | - | - | - | - |
| ATTEMPT* | 96K | 83.8 | 90.0 | 93.1 | 93.7 | 90.8 | 86.1 | 79.9 | 64.3 | 85.2 | 74.4 | 78.3 | 66.5 | 69.2 | 82.1 | 74.1 |
| MPT* | 10.5K | $84.3_{0.57}$ | $90.0_{0.13}$ | $93.0_{0.24}$ | $93.3_{0.26}$ | $90.4_{0.07}$ | $89.2_{0.98}$ | $82.7_{0.41}$ | $63.5_{0.05}$ | $\mathbf{85.8}_{0.14}$ | $74.8_{0.07}$ | $79.2_{0.67}$ | $70.2_{0.82}$ | $67.3_{0.00}$ | $89.3_{0.00}$ | $\mathbf{76.1}_{0.31}$ |

**Table 2:** Results on MRQA and Others. We use F1 for MRQA tasks and accuracy for others as the evaluation metrics. MPT results are averaged over three runs and subscripts indicate standard deviation.

| Method | param/task | MRQA | | | | | Others | | | | |
| --- | --- | --- | --- | --- | --- | --- | --- | --- | --- | --- | --- |
| | | NQ | HP | SQA | News | Avg. | WG | Yelp | SciTail | PAWS | Avg. |
| Finetuning | 220M | 75.1 | 77.5 | 81.1 | 65.2 | **74.7** | 61.9 | 96.7 | 95.8 | 94.1 | **87.1** |
| Adapters | 1.9M | 74.2 | 77.6 | 81.4 | 65.6 | 74.7 | 59.2 | 96.9 | 94.5 | 94.3 | 86.2 |
| BitFit | 280K | 70.7 | 75.5 | 77.7 | 64.1 | 72.0 | 57.2 | 94.7 | 94.7 | 92.0 | 84.7 |
| PT | 76.8K | 67.9 | 72.9 | 75.7 | 61.1 | 69.4 | 49.6 | 95.1 | 87.9 | 55.8 | 72.1 |
| SPoT | 76.8K | 68.2 | 74.8 | 75.3 | 58.2 | 69.1 | 50.4 | 95.4 | 91.2 | 91.1 | 82.0 |
| ATTEMPT | 232K | 70.4 | 75.2 | 77.3 | 62.8 | 71.4 | 57.6 | 96.7 | 93.1 | 92.1 | 84.9 |
| MPT | 77.6K | $72.0_{0.11}$ | $75.8_{0.14}$ | $77.2_{0.05}$ | $63.7_{0.06}$ | $72.2_{0.09}$ | $56.5_{0.87}$ | $96.4_{0.01}$ | $95.5_{0.26}$ | $93.5_{0.13}$ | $85.5_{0.32}$ |

adapt a pretrained model to all the target tasks using multitask learning, such as HyperFomer (Mahabadi et al., 2021), HyperDecoder (Ivison & Peters, 2022), multitask variants of FT and Adapters. We directly quote numbers reported in published papers when possible or use publicly available source code (Karimi Mahabadi et al., 2021; Mahabadi et al., 2021; Asai et al., 2022) under the same backbone and experimental settings for a fair comparison.

**Implementation details.** For source training, we train MPT on the mixture of source tasks for 5 epochs with the examples-proportional mixing strategy (Raffel et al., 2020) and stochastic task sampling described in Section 3.2. For prompt distillation, we calculate the hidden state loss for hidden states from both the encoder and decoder of T5. For target adaptation, we reuse the shared prompt from MPT and take averaged source task-specific vectors to initialize the target task-specific vector. We run all the experiments three times with different random seeds and report the mean and standard deviations. In few-shot experiments, for each number of shots $k$, we randomly sample 10 times from the training set with different random seeds and report the mean performances. Note that for few-shot learning, the source prompt learning still uses the full set of the source tasks. See Appendix A for the full experimental setup including hyperparameters.

## 4.2 RESULTS AND ANALYSIS

**Full-dataset adaptation.** Tables 1 and 2 show the per-task performance of different methods on all four benchmarks. As seen from Table 1 (top), MPT establishes new state-of-the-art results for parameter-efficient finetuning on both GLUE and SuperGLUE. When compared to vanilla PT (Lester et al., 2021), MPT obtains a relative improvement of 13% on GLUE and 16% on Super-GLUE with the same number of task-specific parameters, highlighting the benefits of transferring knowledge from multiple source tasks. MPT also consistently outperforms other parameter-efficient methods such as SPoT (Vu et al., 2022), ATTEMPT (Asai et al., 2022), and BitFit (Zaken et al., 2022), despite updating far fewer parameters. Adapters is the most competitive in terms of average accuracy on both benchmarks, but MPT is far more parameter efficient and requires $4\times$ fewer task-specific parameters. More surprisingly, MPT outperforms the full finetuning baseline on both benchmarks, despite tuning 0.035% as many task-specific parameters. See Figure 2 for the comparison against different methods in terms of accuracy and parameter-efficiency.

Table 1 (bottom) shows the results when finetuning against a *group* of target tasks. ATTEMPT and MPT are particularly performant in this setting, even when compared against state-of-the-art

**Table 3:** Few-shot learning results with $k = \{4, 16, 32\}$ on BoolQ, CB, and SciTail. FT: Finetuning, AD: Adapters, PT: Prompt tuning, ST: SPoT, HF: HyperFormer, ATP: ATTEMPT. Numbers in brackets denote the number of parameters tuned for each task. MPT is very competitive or even better than existing methods in the majority of the cases while tuning much fewer task-specific parameters.

| $k$-shot | | FT (220M) | AD (1.9M) | PT (76.8K) | ST (76.8K) | HF (638K) | ATP (232K) | MPT (77.6K) |
|---|---|---|---|---|---|---|---|---|
| BoolQ | 4 | 50.5 | 53.4 | 61.6 | 50.5 | 48.0 | 61.8 | **62.2** |
| | 16 | 56.5 | 51.4 | 61.9 | 50.6 | 50.2 | 60.0 | **63.3** |
| | 32 | 58.4 | 54.5 | 61.7 | 61.2 | 58.3 | 65.3 | **68.9** |
| CB | 4 | 57.7 | 51.1 | 53.5 | 71.4 | 60.7 | **82.1** | 73.6 |
| | 16 | 77.0 | 74.8 | 63.5 | 64.3 | 76.3 | 78.5 | **78.6** |
| | 32 | 80.0 | 74.8 | 67.8 | 64.3 | 81.4 | **85.7** | 82.1 |
| SciTail | 4 | 79.6 | 79.5 | 57.7 | 69.6 | 82.0 | 80.2 | **80.2** |
| | 16 | 80.0 | 83.2 | 60.8 | 71.9 | 86.5 | 79.5 | **87.3** |
| | 32 | 81.9 | 85.0 | 60.2 | 71.9 | 85.8 | 80.2 | **86.3** |

**Table 4:** Few-shot learning results on GLUE and SuperGLUE for vanilla prompt tuning (PT) and MPT with 4, 16, and 32 training examples. MPT consistently outperforms PT, demonstrating the generalizability of MPT prompts to new tasks with only a few training examples.

| $k$-shot | Method | GLUE | | | | | | | | | SuperGLUE | | | | | |
|---|---|---|---|---|---|---|---|---|---|---|---|---|---|---|---|---|
| | | MNLI | QQP | QNLI | SST-2 | STS-B | MRPC | RTE | CoLA | Avg. | Multi | BoolQ | WiC | WSC | CB | Avg. |
| 4 | PT | 40.1 | 63.2 | 40.4 | 53.0 | 88.8 | 68.1 | 56.3 | 27.4 | 54.7 | 61.8 | 61.6 | 51.2 | 60.4 | 53.5 | 57.7 |
| | MPT | 59.4 | 82.0 | 86.2 | 56.5 | 89.1 | 68.1 | 62.6 | 34.8 | 67.3 | 62.2 | 62.2 | 52.9 | 67.3 | 73.6 | 63.6 |
| 16 | PT | 41.5 | 62.3 | 59.9 | 50.9 | 87.8 | 68.1 | 54.7 | 28.5 | 56.7 | 60.3 | 61.9 | 48.9 | 44.2 | 63.5 | 55.8 |
| | MPT | 61.6 | 84.7 | 90.6 | 63.2 | 89.1 | 70.1 | 64.8 | 32.1 | 69.5 | 64.5 | 63.3 | 49.8 | 67.3 | 78.6 | 64.7 |
| 32 | PT | 37.0 | 62.3 | 56.7 | 50.9 | 87.5 | 68.1 | 54.7 | 23.2 | 55.1 | 59.2 | 61.7 | 52.6 | 67.3 | 67.8 | 61.7 |
| | MPT | 63.6 | 88.5 | 91.0 | 75.9 | 89.7 | 74.5 | 59.7 | 30.8 | 71.7 | 63.3 | 68.9 | 53.9 | 67.3 | 82.1 | 67.1 |

multitask baselines such as HyperFormer (Mahabadi et al., 2021) and HyperDecoder (Ivison & Peters, 2022), which train a single model on different target tasks. This reveals the potential of our MPT to further leverage multitask knowledge on the target side, enabling even more parameter-efficient adaptation of pretrained language models.

Table 2 shows the performance of different methods on the MRQA and Others benchmark. Our approach significantly improves the average performance of PT by $+2.8\%$ on MRQA and $+13.5\%$ on the Others benchmark, while adding only $0.01\%$ more task-specific parameters. Similarly, MPT obtains $85.5\%$ average accuracy on WinoGrande, Yelp, SciTail, and PAWS, outperforming BitFit ($84.7\%$), which updates $10\times$ more task-specific parameters. When we increase the prompt length from 100 to 300, we also found an average improvement of $0.8\%$ on MRQA and $0.6\%$ on Others, closing the gap between MPT and Adapters. While our improvements being highly parameter-efficient are encouraging, the accuracy gap between MPT and the full finetuning is still significant in MRQA, which indicates opportunities for future work in multitask prompt tuning.

**Few-shot adaptation.** Following prior works (Mahabadi et al., 2021; Asai et al., 2022), we first conduct few-shot experiments on BoolQ, CB, and SciTail tasks to measure how the pretrained MPT prompts can be generalized to new tasks with only a few training examples available ($k = 4, 16, 32$). Table 3 shows the results of our approach and other baselines, which includes full finetuning, Adapters, HyperFormer, PT, and SPoT. As can be seen from Table 3, vanilla PT struggles for few-shot adaptation (esp., CB and SciTail), suggesting that randomly initialized prompts are hard to generalize to new tasks with only a few labeled examples. SPoT improves the performance of PT on CB and SciTail tasks, and MPT outperforms both PT and SPoT. We also observe that other methods in Table 3 (Finetuning, Adapters, HyperFormer, and ATTEMPT) have trouble in the few-shot setting. Moreover, Table 4 shows the few-shot learning performance comparison between PT and MPT on all the GLUE and SuperGLUE tasks. As shown in Table 4, we can observe that not only MPT outperforms the vanilla PT by a large margin in most of the datasets, but also MPT can perform very well on many datasets to reach their full-dataset performance with 16 or 32 shots, such as QQP, QNLI, STS-B, and WSC. These results clearly indicate that MPT can effectively use cross-task knowledge in source tasks to target tasks where there are only a few labeled examples.

**Natural language generation tasks.** We next conduct experiments to test whether prompt decomposition learned from source NLU tasks can generalize to target NLG tasks. We transfer the T5-Large prompt trained using 6 diverse source tasks to two NLG tasks: E2E (Novikova et al., 2017) and WebNLG (Gardent et al., 2017). Table 5 shows that our proposed MPT significantly outperforms standard PT (Lester et al., 2021) on both NLG tasks across all the metrics. Our BLEU improvements over PT are $3.03\%$ and $6.25\%$ on E2E and WebNLG tasks respectively, showing the

**Table 5:** Results on NLG tasks. The source prompt decomposition is learned against NLU tasks and adapted to target NLG tasks. MPT consistently outperforms PT on both tasks.

| | E2E | | | | | WebNLG | | |
|---|---|---|---|---|---|---|---|---|
| | BLEU | NIST | METEOR | Rouge-L | CIDEr | BLEU | METEOR | TER ($\downarrow$) |
| PT | 29.11 | 5.00 | 0.343 | 51.50 | 1.72 | 46.02 | 0.37 | 46.89 |
| MPT | 32.14 | 5.35 | 0.363 | 52.88 | 1.86 | 52.27 | 0.40 | 41.36 |

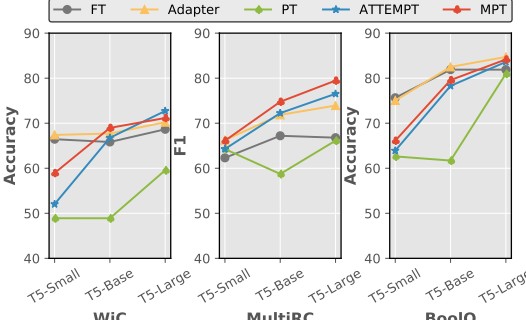 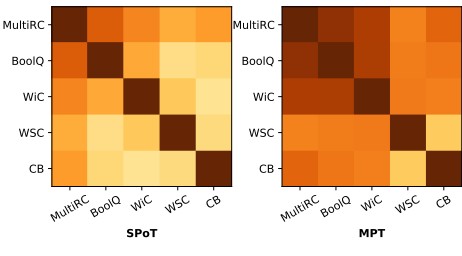

**Figure 4:** (Left) Performance of various baselines as a function of model size (from T5-Small to T5-Large). (Right) Correlation of prompt matrices on SuperGLUE tasks. Best viewed in color.

effectiveness of our approach on both NLU (e.g., classification, NLI, QA tasks) and NLG tasks. This is an impressive result, particularly since the source tasks are all NLU tasks, i.e., MPT can transfer knowledge from NLU tasks to NLG tasks.

**Model scaling.** We conduct scaling experiments to analyze how MPT performs with increasing pretrained model sizes on three SuperGLUE tasks as in Asai et al. (2022). Figure 4 (left) shows the performance of MPT as well as full finetuning (FT), Adapter, PT, and ATTEMPT with three different T5 models (T5-Small, T5-Base, T5-Large). These results show that MPT is not only able to achieve the best parameter efficiency, but also is effective across different model scales ranging from 60M to 770M parameters.

**Analyzing prompt matrices.** We conduct qualitative analyses on prompts learned using MPT to investigate whether cross-task knowledge is indeed encoded in the task-shared prompt, making it easier for target tasks to effectively adapt and encode their own knowledge. Following Vu et al. (2022), we use the prompt matrices to compute cosine similarities between all pairs of target tasks after adaptation, where each task is represented by the composition of task-shared and task-specific prompts (averaged to obtain a single vector). Figure 4 (right) shows the visualization of cosine similarity matrices for SPoT and MPT on SuperGLUE tasks. We find that task embeddings can effectively cluster similar tasks together (e.g., MultiRC is similar to BoolQ).

### 4.3 ABLATION STUDIES

**Prompt decomposition and distillation.** Table 6 presents the results on SuperGLUE where we fix all the hyper-parameters across all settings and rerun MPT source training to get various ablated versions of the transferred prompt.

To measure the effect of prompt decomposition, we replace the vanilla source prompt with our decomposable prompt of task-shared and task-specific components and train it without prompt distillation (third row in Table 6), which gives us 3.5% average performance improvement on SuperGLUE over the baseline (first row in Table 6). This ablation clearly demonstrates the importance of the prompt decomposition strategy in MPT and shows that the shared component can effectively capture the rich cross-task knowledge that is beneficial for target downstream tasks.

**Table 6:** Ablation results on prompt decomposition and distillation.

| Decomposition | Distillation | SuperGLUE Avg. |
|---|---|---|
| ✗ | ✗ | 69.5 |
| ✗ | ✓ | 70.6 |
| ✓ | ✗ | 73.0 |
| ✓ | ✓ | 74.1 |

To test the effect of prompt distillation, we train a vanilla prompt shared by all the source tasks with the same training loss of MPT in Equation 5. The teacher models are kept the same for this ablation and MPT. Compared with the simple baseline (first row in Table 6), adding prompt distillation (second row) produces a 1.1% average performance improvement. Furthermore, we observe that

**Table 7:** MPT performance on MRQA and Others with more source tasks.

| | MRQA | | | | | Others | | | | |
|---|---|---|---|---|---|---|---|---|---|---|
| | NQ | HP | SQA | News | Avg. | WG | Yelp | SciTail | PAWS | Avg. |
| MPT (w/ 6 Source Tasks) | 72.0 | 75.8 | 77.2 | 63.7 | 72.2 | 56.5 | 96.4 | 95.5 | 93.5 | 85.5 |
| MPT (w/ 12 Source Tasks) | 72.1 | 76.4 | 77.9 | 64.0 | 72.6 | 56.6 | 96.8 | 95.9 | 92.9 | 85.6 |

prompt distillation combined with prompt decomposition yields the best average performance of 74.1% on the SuperGLUE benchmark. This confirms that distilling knowledge from separately-trained source prompts is an effective strategy for learning good decomposable prompts.

**Distillation objective.** We further investigate the individual components of prompt distillation to measure their influences on the final performance. We remove the loss of hidden states from Equation 5 and find that it produces an average performance of 73.7% on SuperGLUE, verifying the effectiveness of regularizing hidden states in conjunction with logits to reach its full performance, which is consistent with findings in Sanh et al. (2019). Finally, we consider a variant of distillation loss to match the teacher and student prompts directly by adding an MSE loss to minimize the distance between the two prompts. Replacing our proposed distillation losses with this prompt distance loss and jointly training it with prompt decomposition yield an average SuperGLUE performance of 73.6%, which performs worse than the distillation losses based on logits and hidden states.

**Prompt length.** While our experiments use $l = 100$ prompt vectors, we show in Figure 5 that using longer prompts obtains improvements up to $l = 300$, reaching 76.8% on SuperGLUE. However, further increasing the prompt length from 300 to 400 leads to an absolute 1.8% drop in accuracy, possibly due to overfitting.

**Target adaptation strategy.** When transferring the shared prompt from source to target tasks, we find that only updating task-shared component (i.e., removing task-specific vectors) or only updating task-specific vectors (i.e., freezing task-shared component) produces suboptimal results (62.5% and 71.3% on SuperGLUE). This shows the importance of updating both components (which have different learning rates) for target adaptation.

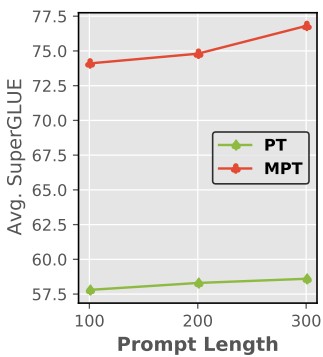

**Figure 5:** Performance on Super-GLUE as a function of prompt length for PT and MPT.

**Stochastic task sampling.** MPT uses a multitask training strategy in Section 3.2, which stochastically samples a number of tasks within each mini-batch. Ablating the stochastic task sampling results in 73.7% on SuperGLUE (lower than the full performance of 74.1%), which demonstrates the slight benefit of this simple multitask training strategy.

**Number of source tasks for pretraining.** For our main experiments, we selected 6 NLP tasks following Asai et al. (2022). To investigate the effect of more source tasks, we incorporated 6 additional diverse source tasks on top of the original 6 tasks, including topic classification (AGNews (Zhang et al., 2015)), multi-choice QA (CommmonsenseQA (Talmor et al., 2019), OpenBookQA (Mihaylov et al., 2018), ARC (Clark et al., 2018)), adversarial NLI (ANLI (Nie et al., 2020)) and commonsense reasoning (Winogrande (Sakaguchi et al., 2021)). Table 7 shows the results on MRQA and Others benchmarks. MPT with 12 tasks is still quite effective for target adaptation on both benchmarks, slightly outperforming MPT trained using 6 tasks. While it is unclear how much MPT would benefit from even more source tasks, it would be interesting to see whether MPT trained on large-scale benchmarks such as CrossFit (Ye et al., 2021)—which consist of 160 NLP tasks—can enable even more parameter-efficient (and accurate) transfer learning.

## 5 CONCLUSION

We introduced and studied multitask prompt tuning (MPT), which learns a single transferable prompt by decomposing and distilling knowledge from multiple source tasks and their task-specific source prompts. MPT decomposes the task prompt as the Hadamard product of a shared prompt matrix and a rank-one task-specific matrix. The shared component is then transferred and adapted to target tasks for further tuning. Empirically we found this approach enables parameter-efficient transfer learning to target downstream tasks across diverse NLP benchmarks, even outperforming the full finetuning baseline in some cases, despite tuning much fewer task-specific parameters.

## ACKNOWLEDGEMENTS

We are grateful to the anonymous reviewers for their constructive comments and suggestions. ZW sincerely thanks Peihao Wang for the insightful discussion during the internship. YK was partially supported by an MIT-IBM Watson AI grant. We also acknowledge support from the IBM Research AI Hardware Center and the Center for Computational Innovation at Rensselaer Polytechnic Institute for the computational resources on the AiMOS Supercomputer.

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

## A EXPERIMENTAL SETUP

For initial training of source prompts, we train MPT on the mixture of source tasks for 5 epochs with the examples-proportional mixing strategy (Raffel et al., 2020) and stochastic task sampling. For prompt distillation, we calculate the hidden state loss for hidden states from both the encoder and decoder of T5. For target adaptation, we reuse the shared prompt from MPT and take averaged source task-specific vectors to initialize the target task-specific vector. We train 20 epochs on small datasets, 10 epochs on large (more than 10k examples) datasets, and 5 epochs on the MRQA datasets. We run all the experiments three times with different random seeds and report the mean and standard deviations. In few-shot experiments, for each number of shots $k$, we randomly sample 10 times from the training set with different random seeds and report the mean performances. For the few-shot setting, the source prompt learning still uses the full set of the source tasks.

During source training, we set the default learning rate as $0.3$ for both task-shared and task-specific components. However, during target adaptation, we use a strategy of two-speed learning rates for those two components, as in Ponti et al. (2022). Specifically, we set the learning rate to $0.3$ and $0.4$ for the task-shared and task-specific components, respectively, during target task adaptation. Following Lester et al. (2021), we set the default number of tunable tokens per each prompt to 100 and initialize the teacher and student prompts by randomly sampling tokens from T5's vocabulary (Raffel et al., 2020). We set the default batch size for T5-Base as 32 and for model scaling experiments, the batch sizes for T5-Small and T5-Large are 100, and 12 respectively. The default input length for most tasks are set to $256$, except MultiRC and MRQA benchmarks have input length of $348$ and $512$. We set the distillation loss coefficient $\lambda$ in Equation 5 to $0.9$ and keep it fixed for all our experiments.

For all datasets, we use the development set as the testing set if the original testing set is not publicly available. If the training set is small, we split the original development set into the development and testing set; otherwise, we separate a development set from the training set and use the original development set for testing. We limit the number of training data for Yelp to 100k.

