# OpenReview forum: "Multitask Prompt Tuning Enables Parameter-Efficient Transfer Learning"
_ICLR.cc/2023/Conference — ICLR 2023 poster_

### Official Review · Reviewer_iVfY · 2022-10-25

**Confidence:** 4
**Clarity, Quality, Novelty And Reproducibility:** Please see the comment above.
**Correctness:** 3
**Technical Novelty And Significance:** 3
**Empirical Novelty And Significance:** 3
**Recommendation:** 6

**Strength And Weaknesses:**

Strengths:
- A novel multitask prompt tuning approach by the separation of shared and task-specific prompt representations and knowledge distillation. The idea is simple and technically sound.
- Evaluations show their approach achieves better results than prior prompt tuning approaches with fewer parameters on widely-adopted benchmarks. Their approach is also on-par or slightly better than the finetuning baseline.
- Their approach demonstrates significantly better few-shot capability than finetune baseline and other prompt tuning approaches.
- Code & data will be released.

Weaknesses:
- Compared to Adapter or finetuning baseline, the proposed approach is still worse on certain datasets (table 1 & 2). It would be better to show if the performance gap w.r.t. Adapter can be closed by adding the same number of prompt parameters as the Adapter.
- It would be more convincing if the few-shot experiments can be performed on GLUE and SuperGLUE, instead of 3 separate datasets. Similar to the scaling experiment (i.e., Figure 4)


**Summary Of The Paper:**

The paper studies multitask prompt tuning to learn better soft prompt representations for tasks. Their proposed approach has two stages: 1) They first train a single source prompt representation for each individual task using the conventional prompt training approach; 2) then they learn a shared prompt representation and task-specific prompt representations on all tasks by applying the prompt distillation from the teacher prompt obtained from step #1. They evaluated their approach on a few well-established benchmarks, including GLUE and SuperGLUE, and demonstrated their prompt tuning approach is better than the prior prompt tuning approaches with less parameters.

**Summary Of The Review:**

The paper proposes a novel multitask prompt tuning approach. Their approach shows better effectiveness and efficiency compared to prior prompt tuning baselines. The results & findings can be more convincing if the weaknesses can be resolved.

---

> ### Author Response · Authors · 2022-11-18
> **Response to Reviewer iVfY (Part 1)**
>
> We thank Reviewer iVfY for acknowledging our approach to be novel and technically sound. Below are our responses to the new experiments and we have incorporated all these changes in the revised version.
>
> (a) **Comparison with Finetuning and Adapter baselines:** Thanks for the great suggestion. Full model finetuning and Adapter are indeed very competitive baselines. While our proposed MPT outperforms both GLUE and SuperGLUE benchmarks (Table 1), full finetuning and adapters still better than MPT (and all other parameter-efficient fine-tuning approaches) on MRQA and Others benchmarks (Table 2). However, they require 2832 and 24 times more parameters than MPT, respectively. While adding the same number of prompt parameters as the Adapter to close the performance gap on MRQA and Others benchmarks is an interesting suggestion, we note that it requires a prompt length of ~2400 tokens on T5-Base, which can be computationally inefficient due to transformer's quadratic complexity with the input length. Following the reviewers suggestion, we increase our prompt length to 100 from 300 and observe an average improvement of 0.8\% on MRQA and 0.6\% on Others, further closing the gap between MPT and Adapters (e.g., only 0.1% difference in Others benchmark: see RTable 1 for individual task performances). We also tested with a prompt length of 400 tokens but did not notice any significant improvements. We believe this is because the optimal prompt length in our current experiments is around 300 tokens, as discussed in our prompt scaling analysis (Section 4.2). Applying MPT for every layer of the pretrained model, instead of the only input layer (like P-Tuning v2 [1]), could be a promising direction to further improve the performance: we leave this as an interesting future work. We have added a discussion on this in Appendix C of the revised manuscript.
>
> RTable 1: Performance on MRQA and Others benchmark by scaling prompt length. All results are based on T5-Base model.
> |             |            | MRQA |      |      |      |      | Others |      |         |      |      |
> |-------------|------------|------|------|------|------|------|--------|------|---------|------|------|
> |             | param/task | NQ   | HP   | SQA  | News | Avg. | WG     | Yelp | SciTail | PAWS | Avg. |
> | Finetuning | 220M       | 75.1 | 77.5 | 81.1 | 65.2 | 74.7 | 61.9   | 96.7 | 95.8    | 94.1 | 87.1 |
> | Adapter     | 1.9M       | 74.2 | 77.6 | 81.4 | 65.6 | 74.7 | 59.2   | 96.9 | 94.5    | 94.3 | 86.2 |
> | MPT-100     | 77.6K      | 72.0 | 75.8 | 77.2 | 63.7 | 72.2 | 56.5   | 96.4 | 95.5    | 93.5 | 85.5 |
> | MPT-300     | 231.5K     | 72.6 | 76.4 | 78.4 | 64.3 | 73.0 | 57.0   | 97.0 | 96.8    | 93.8 | 86.1 |
>
> (b) **Few-shot experiments on GLUE and SuperGLUE:** Thanks for the suggestion! We follow [2] and conduct few-shot experiments on BoolQ, CB, and SciTail tasks for a fair and direct comparison with other parameter-efficient methods, namely SpoT, ATTEMPT and Hyperformer. However, following the reviewer's suggestion, we further conduct more comprehensive few-shot experiments on all the GLUE and SuperGLUE tasks by comparing PT and MPT. As shown in RTable 2, we can observe that MPT outperforms PT by a large margin in most of the datasets. Moreover, MPT can perform very well on many datasets to reach their full-dataset performance with 16 or 32 shots, such as QQP, QNLI, STS-B, and WSC. These results clearly indicate that MPT can effectively use cross-task knowledge in source tasks to target tasks where there are only a few labeled examples. We have added these new results in Appendix D of the revised draft.
>
> RTable 2: Few-Shot results on GLUE and SuperGLUE with k = {4, 16, 32}. MPT consistently outperforms PT by a very large margin, demonstrating generalizability of MPT prompts to new tasks with only a few training examples.
> |    |     | MNLI  | QQP   | QNLI  | SST-2 | STS-B | MRPC  | RTE   | CoLA  | Avg.  | Multi | BoolQ | WiC   | WSC   | CB   | Avg.  |
> |----|-----|-------|-------|-------|-------|-------|-------|-------|-------|-------|-------|-------|-------|-------|------|-------|
> | 4  | PT  | 40.06 | 63.19 | 40.43 | 53.02 | 88.76 | 68.14 | 56.33 | 27.41 | 54.67 | 61.83 | 61.60 | 51.16 | 60.38 | 53.5 | 57.69 |
> |    | MPT | 59.44 | 82.02 | 86.19 | 56.54 | 89.10 | 68.14 | 62.59 | 34.73 | 67.34 | 62.18 | 62.20 | 52.87 | 67.31 | 73.6 | 63.63 |
> | 16 | PT  | 41.54 | 62.31 | 59.87 | 50.92 | 87.78 | 68.14 | 54.68 | 28.53 | 56.72 | 60.32 | 61.90 | 48.90 | 44.23 | 63.5 | 55.77 |
> |    | MPT | 61.63 | 84.68 | 90.66 | 63.15 | 89.05 | 70.10 | 64.75 | 32.10 | 69.52 | 64.45 | 63.30 | 49.84 | 67.31 | 78.6 | 64.70 |
> | 32 | PT  | 37.00 | 62.25 | 56.77 | 50.93 | 87.46 | 68.14 | 54.68 | 23.23 | 55.06 | 59.24 | 61.70 | 52.57 | 67.31 | 67.8 | 61.72 |
> |    | MPT | 63.55 | 88.46 | 91.03 | 75.92 | 89.71 | 74.51 | 59.71 | 30.82 | 71.71 | 63.25 | 68.90 | 53.92 | 67.31 | 82.1 | 67.10 |

---

> > ### Author Response · Authors · 2022-11-18
> > **Response to Reviewer iVfY (Part 2)**
> >
> > **References:**
> >
> > [1] Xiao Liu, Kaixuan Ji, Yicheng Fu, Weng Lam Tam, Zhengxiao Du, Zhilin Yang, Jie Tang. P-Tuning v2: Prompt Tuning Can Be Comparable to Fine-tuning Universally Across Scales and Tasks. ACL 2022.
> >
> > [2] Akari Asai, Mohammadreza Salehi, Matthew E Peters, and Hannaneh Hajishirzi. Attentional mixtures of soft prompt tuning for parameter-efficient multi-task knowledge sharing. EMNLP, 2022.

---

### Official Review · Reviewer_6YdE · 2022-10-25

**Confidence:** 4
**Correctness:** 3
**Technical Novelty And Significance:** 3
**Empirical Novelty And Significance:** 3
**Recommendation:** 6

**Clarity, Quality, Novelty And Reproducibility:**

- The paper is clearly presented;
- Though multi-task prompting in NLP has been investigated in [2,3], distillation [4] decomposition of prompting has been investigated in [4], the idea to use low-rank decomposition for multi-task target prompt adaptation is interesting and new.
- Some implementation details regarding the might also be vital for reproducibility.


[2] Vu, Tu, et al. "SPoT: Better Frozen Model Adaptation through Soft Prompt Transfer." Proceedings of the 60th Annual Meeting of the Association for Computational Linguistics (Volume 1: Long Papers). 2022.
[3] Sanh, Victor, et al. "Multitask prompted training enables zero-shot task generalization." ICLR (2022).
[4] Zhong, Qihuang, et al. "Panda: Prompt transfer meets knowledge distillation for efficient model adaptation." arXiv preprint arXiv:2208.10160 (2022).

**Strength And Weaknesses:**

Summary Of Strengths
- the paper is clearly written and presented;
- the idea of leveraging decomposition is new and insightful, which not only makes the prompt learning more performant but results in fewer parameters.
- extensive ablations are provided (decomposition, distillation, adaptation strategy, and training strategy) to illustrate the design choices, which may pave the road for future researchers and practitioners in the prompt learning area.
- the efficacy of the decomposition and distillation for multi-task prompt tuning is verified across benchmarks (GLUE, SuperGLUE, MRQA) and scales (up to 700M);

Summary Of Weaknesses
- the additional training compute is unclear compared to fine-tuning, is the fine-tuning also conducted using the same schedule as MPT; Also, in the SPoT paper, they found the best results were achieved with multi-task fine-tuning 79.2 (T5-base), is it also a valid baseline to compare with?
- the poor baseline performance

   (i) It seems the baseline of BitFit / LoRA / Adapter performs worse than the one reported in [1], could the author elaborate on the reasons?

   (ii) Also, is there any intuition  why SPoT yields such worse performance on SuperGLUE, which should be comparable to Model Tuning in the original paper (though they use more source tasks compared to the one used here)

- the adaptation in Table 1 and 2 is still per-task adaption but not in a multi-task manner (the design choice of this is unclearly presented), and how to select the group is unclearly presented;

- the generalization of the proposed method is uncertain, is it limited to T5-variants, or is it also applicable to GPT (casual mask) models?

- will the variance of different runs also be given in Table 1 and 2, which can help show the significance of the results?

- for the few-shot setting, is the source prompt learning still using the full set of the source tasks, or is that also few-shot?

[1] Sung, Yi-Lin, Jaemin Cho, and Mohit Bansal. "Lst: Ladder side-tuning for parameter and memory efficient transfer learning." NeurIPS (2022).


**Summary Of The Paper:**

This paper proposed a new method for multi-task prompt tuning, which uses source tasks to learn a single shared prompt and then adapts to target tasks with decomposition and distillation. The design of the decomposition makes the resulting prompting learning more performant yet more parameter-efficient.

**Summary Of The Review:**

See above.

---

> ### Author Response · Authors · 2022-11-18
> **Response to Reviewer 6YdE (Part 1)**
>
> We thank the Reviewer 6YdE for acknowledging that our idea to use low-rank decomposition for multi-task target prompt adaptation is interesting and new. Below we address the reviewer's concerns and have incorporated all the feedback in the revised draft.
>
> (a) **Training cost compared with fine-tuning:** Similar to existing works on prompt transfer (SPoT [1] and ATTEMPT [2]), our proposed MPT consists of two training stages, source training and target adaptation (explained in Section 3.2). Traditional model fine-tuning directly trains on the target downstream task, which shares the exact same target training schedule as our MPT, although ours is far more parameter efficient than model fine-tuning (220M vs 77.6K: i.e., only tuning 0.035% as many task-specific parameters). The additional training compute of MPT is the source training for learning a single transferable prompt by decomposing and distilling knowledge from multiple task-specific source prompts. Considering the significant performance improvement the source training brings, this additional training cost seems to be an acceptable trade-off to make for parameter-efficient transfer learning, as in [1, 2]. More importantly, its computation overhead can be amortized, since we only need to conduct the source training **once** such that the transferrable prompt can be adapted into many target task afterwards. In addition, our prompt decomposition leverages low-rank updates to task-specific components that introduce a minimal amount of computation, which further reduces the cost of source training.
>
> (b) **Baseline of multitask fine-tuning from SPoT [2]:** We have included multi-task fine-tuning methods as one of the baselines in Table 1 (second part of the table), where all methods (with the * symbol) take multiple target tasks as input and perform multi-task learning on the groups of GLUE and SuperGLUE separately. We adopt their numbers directly from HyperFormer [3], HyperDecoder [4] and ATTEMPT [2]. Now, regarding the performance of 79.2 (T5-Base) from SPoT, we would like to point out the difference in backbone LMs. In particular, the multi-task finetuning baseline in SpoT uses T5 v1.1 (which was pretrained exclusively on span corruption), while we adopt T5 as the backbone LM for all our experiments (following much prior work, e.g. LST [5], HyperFormer [3], Compacter [6]).
>
> \(c\) **Poor baseline performances:**
> - *(1) Performance difference of BitFit / LoRA / Adapter from LST [5]:* Thanks for pointing us to the very recent reference. We adopt the numbers of BitFit / Adapter directly from ATTEMPT [2] (LoRA is not included). However, following reviewer's suggestion, we carefully checked the LST paper and find that both LST and ATTEMPT reproduce BitFit and Adapter based on the same codebase of Compacter [6], and their performance differences can be explained by the following reasons. First, LST increases the number of parameters of Adapter (1.63% parameters of T5-Base), while the Adapter from ATTEMPT and Compacter updates 0.832% parameters of T5-Base. This explains why LST's Adapter performs slightly better than our Adapter.  Second, LST reports the average of F1 score and accuracy on MRPC, while ATTEMPT only reports the accuracy. We confirmed that the F1 score is higher than the accuracy on MRPC. Finally, LST reports the average result over three runs, while ATTEMPT only reports one single run, which explains the performance variance on CoLA (a very unstable task).
> - *(2) Performance of SpoT on SuperGLUE:* This is due to two main reasons: (i) SpoT follows the original prompt tuning paper [7] and use uses T5 v1.1 LM-adapt as the backbone LMs, which is different from T5 model used in our work including ATTEMPT and others; (ii) SpoT uses significantly more source tasks compared to our setup, as rightly pointed by the reviewer. As discussed in [2], T5-LM adapt v1.1 is especially very sensitive and hard to tune while using it as a backbone LM for parameter-efficient approaches. To summarize, all the baselines including ours use T5 as backbone LMs, making our baseline comparisons fair across all the benchmarks.

---

> > ### Author Response · Authors · 2022-11-18
> > **Response to Reviewer 6YdE (Part 2)**
> >
> > (d) **Generalization of MPT:** Thanks for this great question. Our current MPT is built on top of prompt tuning [7], which is mostly applied to T5. So, we follow prior works, such as SPoT [1] and ATTEMPT [2], to conduct experiments on T5-variants. However, our proposed approach is quite generic and can be applied to both T5 and GPT models. This is primarily because MPT only prepends a prompt matrix (i.e., virtual tokens) to the input embedding layer and hence can be adopted to any transformer models (encoder-only, encoder-decoder, or decoder-only), not limited to T5. Specifically, MPT focuses on decomposing the prompt matrix into task-specific and task-shared components, which introduces minimal intrusion to the backbone model. Similarly, the distillation part of MPT is also model-agnostic and can be generalized to GPT models. We leave the extension of MPT to GPT models as an interesting future work. We have added this discussion in Appendix E of the revised paper.
> >
> > (e) **Variance in Table 1 and 2:** We run all our experiments three times with different random seeds and report the mean numbers. Following reviewer’s suggestion, we have now added the standard deviation of our results in Table 1 and Table 2. For baseline numbers adopted from published papers such as ATTEMPT and HyperFormer, they do not report standard deviations.
> >
> > (f) **Source prompt learning for few-shot setting:** Yes, for the few-shot setting, the source prompt learning still uses the full set of the source tasks. Few-shot is only applied to target adaptation.
> >
> > (g) **Difference with existing works:** We thank the reviewer for pointing out these relevant papers, especially, multi-task prompting [1, 8] and distillation [9]. First, we are glad to see more active ongoing research investigating the general ideas of multi-task learning and distillation, indicating their importance and potential. Second, as pointed by the reviewer 6YdE (and reviwer FfyU), our unique novelty lies in combining low-rank decomposition and distillation to enable efficient multi-task prompt learning and transfer, which has key differences with existing works. More specifically, multi-tasking in T0 [1] retrains the whole T5 model straightforwardly by multi-task multi-prompt learning to enable zero-shot performance, unlike the problem of soft prompt transfer we consider in our work. While SPoT [8] does have a baseline of multi-task training of prompts, we find simply training a single soft prompt by simply mixing all task knowledge into a joint parameter space is sub-optimal as it fails to leverage commonalities across source tasks while minimizing interference. In contrast, our low-rank decomposition separates task-specific information to learn better task-shared knowledge for effective parameter efficient adaptation on target tasks. Lastly, the concurrent work PANDA [9] uses distillation with a new metric to better predict transferability across different combinations of source-target tasks. This significantly differs from MPT, which uses a low-rank prompt decomposition to leverage commonalities across the source tasks while minimizing interference between them. In addition, PANDA focuses on transferring from one source task to another target task using a similarity measure, while MPT leverages multitask learning to better exploit the rich cross-task knowledge in prompt transfer.
> >
> > (h) **Reproducibility.** We will publicly release our code and trained prompts to facilitate reproducibility.
> >
> >
> > **References:**
> >
> > [1] Tu Vu, Brian Lester, Noah Constant, Rami Al-Rfou, and Daniel Cer. Spot: Better frozen model adaptation through soft prompt transfer. ACL, 2022.
> >
> > [2] Akari Asai, Mohammadreza Salehi, Matthew E Peters, and Hannaneh Hajishirzi. Attentional mixtures of soft prompt tuning for parameter-efficient multi-task knowledge sharing. EMNLP, 2022.
> >
> > [3] Rabeeh Karimi Mahabadi, Sebastian Ruder, Mostafa Dehghani, and James Henderson. Parameter-efficient multi-task fine-tuning for transformers via shared hypernetworks. ACL, 2021.
> >
> > [4] Hamish Ivison and Matthew E Peters. Hyperdecoders: Instance-specific decoders for multi-task nlp. Findings of EMNLP, 2022.
> >
> > [5] Yi-Lin Sung, Jaemin Cho, and Mohit Bansal. LST: Ladder Side-Tuning for Parameter and Memory Efficient Transfer Learning. NeurIPS, 2022.
> >
> > [6] Rabeeh Karimi Mahabadi, James Henderson, and Sebastian Ruder. Compacter: Efficient low-rank hypercomplex adapter layers. NeurIPS, 2021.
> >
> > [7] Brian Lester, Rami Al-Rfou, and Noah Constant. The power of scale for parameter-efficient prompt tuning. EMNLP, 2021.
> >
> > [8] Victor Sanh, Albert Webson, Colin Raffel, Stephen Bach, Lintang Sutawika, Zaid Alyafeai, Antoine Chaffin, Arnaud Stiegler, Teven Le Scao, Arun Raja, et al. Multitask prompted training enables zero-shot task generalization. ICLR, 2022.
> >
> > [9] Qihuang Zhong, Liang Ding, Juhua Liu, Bo Du, and Dacheng Tao. Panda: Prompt transfer meets knowledge distillation for efficient model adaptation. arXiv preprint arXiv:2208.10160, 2022.

---

### Official Review · Reviewer_FfyU · 2022-10-26

**Confidence:** 4
**Correctness:** 4
**Technical Novelty And Significance:** 3
**Empirical Novelty And Significance:** 3
**Recommendation:** 8

**Clarity, Quality, Novelty And Reproducibility:**

- The paper is very well written, with clear notation, figures and exposition.
- Overall novelty is modest, but the method is simple and provides benefits
- Authors will provide code for reproducible results

**Strength And Weaknesses:**

Strengths
- Parameter-efficient transfer learning is an important research area
- The prompt decomposition method is quite straightforward (decomposition + distillation)
- Comprehensive evaluation (21 datasets) and baseline methods
- Nice breadth of additional experiments (few-shot performance, LM param scaling, prompt lenght)
- Ablation studies highlight benefits of combining decomposition + distillation

Weaknesses
- Ideally the manuscript would explore some class of larger language models (3 - 11B param range), though this presupposes some level of compute that is not available to all researchers, so it is not a strong criticism.
- Experiments would benefit from replicates to characterize variance.
- The core methods aren't super novel (decomposition + distillation), but the combination seems to provide empirical benefits.
- Code is not immediately available.

**Summary Of The Paper:**

The paper presents a soft (continuous) prompt tuning method called MPT. In traditional soft prompt tuning, prompts are often sensitive to initialization when trained from scratch and performance may still lag behind full model fine tuning. In this work, the manuscript presents a method for multitask prompt tuning where a single soft prompt is learned that can be transferred to target tasks. The authors find a low rank decomposition based on a source task matrix and a task-specific low rank matrix is more performative than sharing the prompts directly across tasks. This decomposition is is learned via a knowledge distillation style approach.

The authors evaluate performance on 21 NLP datasets reflecting a variety of tasks and report significant improvements on SuperGLUE vs vanilla prompt tuning, along with all the smaller training parameter benefits of parameter efficient transfer learning. They further find that MPT performs well in few-shot learning for models in the 60M to 770M parameter space. The paper presents comprehensive ablation experiments

**Summary Of The Review:**

I think this paper makes several nice empirical contributions, is very clearly written with comprehensive evaluations and ablation experiments. This covers some reasonable questions in the space of soft prompt tuning and MTL so it merits acceptance.

---

> ### Author Response · Authors · 2022-11-18
> **Response to Reviewer FfyU**
>
> We thank Reviewer FfyU for the positive recommendation and constructive comments. Below are our responses on the larger language models and technical novelty. We have incorporated all the feedback and suggestions in the revised paper.
>
> (a) **MPT with billion+ parameter language models:** Thanks so much for the suggestion and your understanding of the compute constraints. While we currently do not possess the compute resources for exploring very large language models with billion+ parameters, we agree with the reviewer that it would be interesting to study the extent to which the benefits of MPT remain at the scale of models like T5-3B and T5-11B. We hope to cover this in our future work.
>
> (b) **Variance of MPT:** Thanks for the suggestion. We run all our experiments three times with different random seeds and report the mean numbers. Following reviewer's suggestion, we have now added standard deviation of our results in Table 1 and Table 2. For baseline numbers adopted from published papers such as ATTEMPT and HyperFormer, they don’t have the variance reported.
>
> \(c\) **Novelty:** The core contributions of our work comes from the novel approach in prompt tuning and strong empirical evidence on a diverse set of benchmarks. We believe our idea of learning a single transferable prompt by decomposing and distilling knowledge from task-specific source prompts is unique, which not only makes the prompt learning more performant but results in fewer parameters.
>
> (d) **Code:** All our code and models will be made publicly available.

---

### Official Review · Reviewer_Tz5t · 2022-11-04

**Confidence:** 4
**Correctness:** 3
**Technical Novelty And Significance:** 3
**Empirical Novelty And Significance:** 3
**Recommendation:** 6

**Clarity, Quality, Novelty And Reproducibility:**

As shown in Fig 5, it seems that 300 is still not the optimal prompt length for MPT. Can you use larger lengths and find the optimal length (the one with the best performance)?



**Strength And Weaknesses:**

## Strength

- A novel method for multi-task prompt tuning. The design of the prompt decomposition and distillation is intuitive and reasonable.
- Great empirical results on GLUE and SuperGLUE. The proposed method MPT outperforms many recent baseline methods for prompt tuning.
- Comprehensive analysis with ablation studies and qualitative analysis with heat maps.

## Weakness
- The evaluation does not consider the NLG tasks, such as those in the GEM benchmark. This can be a big limitation.
- The evaluation is based on the setup where only 6 source tasks are used. This is a pretty small size and it seems that many source tasks are closely related to each other. I would suggest authors use benchmarks such as CrossFit (Ye et al. 2022) to do a more large-scale analysis, where the transferring is more challenging as some source tasks can be relatively less related to the target tasks.
- The current method design only considers a single shared prompt for transfer. I think when you have a large number of source tasks, this can be a weak point. As it is less likely that a very diverse set of tasks can use a single prompt to share all knowledge.

**Summary Of The Paper:**

This paper focuses on the prompt tuning of Transformer language models in multi-task settings. They proposed a method named multitask prompt tuning (MPT), which aims to enhance the transferability of source prompts (i.e., the learned soft prompts for source tasks). Specifically, they learn a single transferable prompt by knowledge distillation from multiple source tasks and then learn the rank-one matrix for adapting the transferable shared prompt to a given target task -- prompt decomposition, distillation, and adaptation. On the benchmark GLUE and SuperGLUE, they compare the proposed MPT method with many other prompt tuning baselines. The MPT outperforms the baseline methods and has a smaller number of the parameters.

**Summary Of The Review:**

Overall, I enjoy reading the paper. The idea is pretty novel and its performance is very good, especially when we consider the parameter efficiency. It has a few limitations which are not stated and covered, though. Also, I think the evaluation can be further improved according to my above suggestions.

---

> ### Author Response · Authors · 2022-11-18
> **Response to Reviewer Tz5t (Part 1)**
>
> We thank the reviewer for the insightful questions and great suggestions. We have revised the paper by including new experiments on NLG tasks, more source tasks and optimal prompt length.
>
> (a) **MPT for NLG tasks:** Thanks for this great suggestion! Our approach falls into parameter-efficient fine-tuning approaches with a rich line of previous works, including Prompt tuning, SPoT, ATTEMPT, HyperFormer, Hyperdecoder, and Compacter, etc. We follow the evaluation protocol of them to conduct our experiments on GLUE, SuperGLUE, MRQA and Other benchmarks. However, as suggested by the reviewer, we performed new experiments on NLG tasks by applying T5-MPT source prompt on target NLG tasks. In particular, we transfer the T5-Large prompt trained using 6 diverse source tasks used in our current experiments for adaptation to two target data-to-text generation tasks, namely E2E [1] and WebNLG [2].
>
> RTable 1: Applying MPT-T5-Large prompts to NLG tasks.
> |             | E2E   |      |        |         |       | WebNLG |        |       |
> |-------------|-------|------|--------|---------|-------|--------|--------|-------|
> |             | BLEU  | NIST | METEOR | R-L | CIDEr | BLEU   | METEOR | TER&darr;  |
> | PT          | 29.11 | 5.00 | 0.343  | 51.50   | 1.72  | 46.02  | 0.37   | 46.89 |
> | MPT         | 32.14 | 5.35 | 0.363  | 52.88   | 1.86  | 52.27  | 0.40   | 41.36 |
>
> RTable 1 shows that MPT significantly outperforms standard PT on both NLG tasks across all the metrics. Our BLEU improvements over PT are 3.03% and 6.25% on E2E and WebNLG tasks respectively, showing the effectiveness of our approach on both NLU (e.g., classification, NLI, QA tasks) and NLG tasks. This is particularly an impressive result since the source tasks were all NLU tasks, i.e., MPT can transfer knowledge from NLU tasks to NLG tasks! We have added this result in the updated version (see Table 5 in Appendix A of the revised paper).
>
> (b) **MPT with more source tasks:** Thanks for the constructive comment. We follow prior published work, ATTEMPT [3] and select datasets with more than or around 100k annotations as source tasks. They are 6 representative NLP tasks including 2 NLI, 1 paraphrase, 1 sentiment analysis and 2 large-scale QA tasks, which are general/diverse enough and can enable knowledge transfer to other tasks. We also consider 21 diverse target tasks consisting of 4 different benchmarks, where some source and target tasks are distantly related (e.g., tasks from Others benchmark are from very different domains compared to source tasks).
>
> However, we think the reviewer's suggestion is very interesting (i.e., adding more remotely relevant source task), and thus investigated this setting by adding 6 additional diverse source tasks, including one topic classification (AGNews), three multi-choice QA (CommmonsenseQA, OpenBookQA, ARC), one adversarial NLI (ANLI) and one commonsense (winogrande) dataset. RTable 2 shows the results on MRQA and Others benchmarks. As can be seen, MPT with 12 more diverse source tasks is still very effective for target adaptation on both benchmarks, slightly outperforming MPT trained using 6 tasks. We have added this additional result in Appendix B of the revised draft.
>
> RTable 2: MPT performance on MRQA and Others with more number of source tasks.
> |        | MRQA  |       |       |       |       | Others |       |         |       |       |
> |--------|-------|-------|-------|-------|-------|--------|-------|---------|-------|-------|
> |        | NQ    | HP    | SQA   | News  | Avg.  | WG     | Yelp  | SciTail | PAWS  | Avg.  |
> | MPT (w/ 6 source tasks)  | 72.0 | 75.8 | 77.2 | 63.7 | 72.2 | 56.5  | 96.4 | 95.5   | 93.5 | 85.5 |
> | MPT (w/ 12 source tasks) | 72.1 | 76.4 | 77.9 | 64.0 | 72.6 | 56.6  | 96.8 | 95.9   | 92.9 | 85.6 |
>
> Finally, we agree with the reviewer that it would be compelling to use benchmarks like CrossFit [4] consisting of 160 NLP tasks as source tasks for analyzing the performance of MPT on parameter-efficient transfer learning. While we currently do not possess the compute resources for this extreme large-scale study (+ the short rebuttal time window), we hope to cover this an interesting future work. Last but not least, we will release pretrained source task prompts and easily extendable code to motivate further studies on task scaling and understanding task transferability across a more diverse set of source and target tasks.

---

> > ### Author Response · Authors · 2022-11-18
> > **Response to Reviewer Tz5t (Part 2)**
> >
> > \(c\) **Single prompt for transfer:** While a single shared prompt enables highly parameter-efficient adaptation to target tasks, we believe the use of a very large diverse set of source tasks may require deeper prompting via adding prompts to every layer of the pretrained model (as in P-Tuning v2 [5]), to superimpose all the tasks into a single multitask prompt. Another potential solution is to first group/cluster source tasks and then apply MPT to each group separately instead of considering all of them together. Finally, given a target task, one can adopt an attention mechanism to combine all MPT prompts to improve both efficiency and task performance: we leave this as an interesting topic for future work.
> >
> > (d) **Optimal prompt length:** Following reviewer's suggestion, we increase the prompt length to 400 and test on SuperGLUE tasks. While varying prompt length, we noticed an average improvement of 2.7% when prompt length is increased from 100 to 300 (74.1 vs 76.8) on SuperGLUE. However, further increase of prompt length from 300 to 400 lead to 1.8% drop in accuracy (76.8 vs 75.0), indicating the optimal prompt length to be of 300 in our experiments.
> >
> > **References:**
> >
> > [1] Jekaterina Novikova, Ondřej Dušek, and Verena Rieser. The E2E dataset: New challenges for end-to-end generation. arXiv preprint arXiv:1706.09254, 2017.
> >
> > [2] Claire Gardent, Anastasia Shimorina, Shashi Narayan, and Laura Perez-Beltrachini. The WebNLG challenge: Generating text from RDF data. ICNL, 2017.
> >
> > [3] Akari Asai, Mohammadreza Salehi, Matthew E Peters, and Hannaneh Hajishirzi. Attentional mixtures of soft prompt tuning for parameter efficient multi-task knowledge sharing. EMNLP, 2022.
> >
> > [4] Qinyuan Ye, Bill Yuchen Lin, and Xiang Ren. CROSSFIT: A Few-shot Learning Challenge for Cross-task Generalization in NLP. EMNLP 2021.
> >
> > [5] Xiao Liu, Kaixuan Ji, Yicheng Fu, Weng Lam Tam, Zhengxiao Du, Zhilin Yang, Jie Tang. P-Tuning v2: Prompt Tuning Can Be Comparable to Fine-tuning Universally Across Scales and Tasks. ACL 2022.

---

### Author Response · Authors · 2022-11-18
**Summary of Authors' Response**

We would like to thank all the reviewers for their constructive comments! We are encouraged to see that reviewers thought that: (a) our design of the prompt decomposition and distillation is novel, intuitive, technically sound (R-Tz5t, R-FfyU, R-iVfY) and insightful, which not only makes the prompt learning more performant but results in fewer parameters (R-6YdE); (b) our approach had great empirical results on GLUE and SuperGLUE, outperforming many recent baseline methods for prompt tuning (R-Tz5t), with a nice breadth of additional experiments on few-shot performance and scaling (R-FfyU); \(c\) our experiments on 21 datasets is comprehensive with qualitative analysis and ablation studies (R-Tz5t, R-FfyU), which may pave the road for future researchers and practitioners in the prompt learning area (R-6YdE).

We have addressed all the questions that the reviewers posed with additional experiment comparisons and clarifications. All of these additional experiments and suggestions have been added into the updated PDF. Below, we summarize the main changes to the paper and request the reviewers to take a look at the new additions.

- Additional results on MPT for NLG tasks, as suggested by R-Tz5t,
- Discussion on MPT with more source tasks, as suggested by R-Tz5t,
- Experiments on optimal prompt length, as suggested by R-Tz5t,
- Variance of our results, as suggested by R-FfyU and R-6YdE,
- Discussion on baseline performances and differences with existing works, as suggested by R-6YdE,
- MPT performance on MRQA and Others benchmarks by increasing prompt parameters, as suggested by R-iVfY,
- Few-shot experiments on GLUE and SuperGLUE, as suggested by R-iVfY.

---

### Public Comment · ~Haeju_Lee1 · 2023-03-03
**Plan for code release**

Hi, and thanks for your nice work!

Could you make the code for the paper public?

---

> ### Public Comment · ~Haeju_Lee1 · 2023-03-24
> **Rough plan**
>
> Hi, I would really appreciate it if you have a rough timeline for releasing the code.
>
> Thank you.

---

### Decision · Program_Chairs · 2023-01-20

**Decision:**

Accept: poster

**Justification For Why Not Higher Score:**

Need larger scale experiments (not necessarily bigger models, but more tasks > 100). I know the authors have attempted to run with more tasks (12 instead of 6) following the suggestion from a reviewer. I also understand that there is a compute resource limitation on the authors side. However, since the model is pitched as a multitask model, I still think this kind of experiments is needed to for a higher score.

**Justification For Why Not Lower Score:**

Interesting approach that has the potential to be widely adopted.

**Metareview: Summary, Strengths And Weaknesses:**

This paper presents a novel prompt tuning approach that learns a transferable prompt by distilling from multiple task-specific prompts. The approach is interesting and the experiments verified the efficacy of the method. All reviewers agree that this paper should be accepted.

**Note From Pc:**

if the above contains the word "oral" or "spotlight" please see: "oral" presentation means -> notable-top-5% and "spotlight" means -> notable-top-25%. As stated in our emails, we are disassociating presentation type from AC recommendations